# De novo design of an intercellular signaling toolbox for multi-channel cell–cell communication and biological computation

Pei Du[1,6], Huiwei Zhao[1,2,6], Haoqian Zhang [3,4,6], Ruisha Wang[1,2], Jianyi Huang[1,2], Ye Tian[1], Xudong Luo[1], Xunxun Luo[1,2], Min Wang[1], Yanhui Xiang[5], Long Qian [4], Yihua Chen [1,2], Yong Tao [1,2✉] & Chunbo Lou[2,5✉]

Intercellular signaling is indispensable for single cells to form complex biological structures, such as biofilms, tissues and organs. The genetic tools available for engineering intercellular signaling, however, are quite limited. Here we exploit the chemical diversity of biological small molecules to de novo design a genetic toolbox for high-performance, multi-channel cell–cell communications and biological computations. By biosynthetic pathway design for signal molecules, rational engineering of sensing promoters and directed evolution of sensing transcription factors, we obtain six cell–cell signaling channels in bacteria with orthogonality far exceeding the conventional quorum sensing systems and successfully transfer some of them into yeast and human cells. For demonstration, they are applied in cell consortia to generate bacterial colony-patterns using up to four signaling channels simultaneously and to implement distributed bio-computation containing seven different strains as basic units. This intercellular signaling toolbox paves the way for engineering complex multicellularity including artificial ecosystems and smart tissues.

[1] CAS Key Laboratory of Microbial, Physiological, and Metabolic Engineering and Institute of Microbiology, State Key Laboratory of Microbial Resources, Institute of Microbiology, Chinese Academy of Sciences, Beijing 100101, China. [2] College of Life Science, University of Chinese Academy of Science, Beijing 100149, China. [3] Bluepha Co., Ltd, ZGC Science Park, Changping, Beijing 102206, China. [4] Center for Quantitative Biology, Peking University, Beijing 100871, China. [5] CAS Key Laboratory of Quantitative Engineering Biology, Guangdong Provincial Key Laboratory of Synthetic Genomics, Shenzhen Key Laboratory of Synthetic Genomics, Shenzhen Institute of Synthetic Biology, Shenzhen Institutes of Advanced Technology, Chinese Academy of Sciences, 1068 Xueyuan Avenue, University Town, Nanshan, Shenzhen 518055, China. [6] These authors contributed equally: Pei Du, Huiwei Zhao, Haoqian Zhang. ✉email: taoyong@im.ac.cn; louchunbo@gmail.com

ntercellular signaling is essential for single cells to acquire multicellular behaviors by facilitating division of labor, coordinating population physiological activities, and organizing tissue development and differentiation[1]. The natural gene pool contains a plethora of intercellular communication systems[2,3]. One well-studied case is the bacterial quorum sensing (QS) systems that govern the physiological transition of bacterial populations to form biofilms[4], as well as to express bioluminescence and virulence factors[5]. In multicellular organisms, short- (autocrine), medium- (paracrine), and long- (endocrine) range intercellular signaling is key for the control of spatial and temporal development, generation of immune responses, and maintenance of physiological homeostasis[6]. In analogy to electronic wires that coordinate the large number of computational units in a computer, intercellular signaling systems are chemical wires for a multicellular body to achieve organism-level performance.

Current efforts of engineering complex biological computations in living cells have met with much frustration. This is largely due to our very limited ability to program large-scale genetic circuits that are often resource-taxing and error-prone in a single cell. Taking a divide-and-conquer strategy by packaging computation modules into different cells and wiring them together may break the bottleneck by achieving stability, programmability, and ultimately computational complexity at the cell consortium level[7]. Proof-of-principle studies have included engineered biological spatial patterns[8,9], synthetic microbial ecosystems[10], synchronized genetic oscillators[11], mammalian bio-computers with complexity up to full adder logics[12], therapeutic circuits for antibiotic-free pathogen control[13,14], and autonomous induction systems for metabolic production[15–19]. In most of these studies, communications between different computing units were channeled by the abovementioned QS systems[10,20], in which the signal molecules, acyl-homoserine lactones (AHLs), are synthesized from S-adenosyl-methionine and acyl-Acyl Carrier Proteins (ACPs), and secreted by sender cells, before they are sensed by the corresponding allosteric transcription factors (aTFs) in receiver cells[21]. Beyond AHLs, yeast peptide-pheromones, human histamine and dopamine hormones, and other endogenous signal molecules were also used for synthetic cell–cell communications[19,22–24].

Although natural intercellular signaling systems constitute a huge repertoire of genetic materials for the engineering of multicellular bio-computation, two aspects limit their applicability. Universality: ideal cell–cell communications should work in a modular manner applicable to a wide range of cell types, especially for scenarios requiring cross-kingdom communications such as microbiome therapy. However, intercellular signaling systems used in previous studies either required the addition of exogenous precursors to synthesize signal molecules or were mechanistically incapable of being transferred from one species to another[12,23,25,26]. Orthogonality: ideal cell–cell communications rely on an array of well-insulated channels for correct signaling. In electronics, insulation of different channels is usually achieved by spatial segregation, whereas in biological systems the most feasible way to achieve insulation is through chemical orthogonality. Recent studies have quantitatively revealed the extensive cross-talk among the conventionally used QS systems[20,27–29], which can be largely attributed to the structural similarity among AHLs and among the corresponding aTFs. To eliminate cross-talk, a number of strategies have been attempted, including rational engineering of the signal-sensing promoters[20], directed evolution of signal-sensing aTFs[30], and large-scale screening of kinase–substrate pairs[31].

We aim to design a truly modular intercellular signaling toolbox for multi-channel cell–cell communications and biological computations by targeting precisely these two key

properties. Specifically, universality is achieved by choosing universal cellular metabolites as precursors for synthesizing the selected small molecules as the signal molecules and designing minimal biosynthetic pathways from the common precursors, and orthogonality is achieved by taking advantage of the chemical diversity of biologically synthesized small molecules and the abundant resource of small molecule-sensing aTFs. Taking a de novo approach combining biosynthetic pathway design, genetic circuit engineering, and directed evolution, we have designed ten novel intercellular signaling systems as cell–cell communication channels, of which six are successfully obtained and quantitatively characterized in *Escherichia coli*. Subsequently, two of them are transferred to yeast *Saccharomyces cerevisiae* and one to human HEK-293T cells for cross-kingdom communication. To demonstrate the advantage of the intercellular signaling toolbox, genetic circuits operating multi-channel (two-, three-, and four-channel) communications are constructed to form biological spatial patterns and to implement an AND–XOR function by coordinating seven NOR/Buffer gate cells. We believe this intercellular signaling toolbox would significantly expand the capability of synthetic biology in multicellular organism engineering and present one of the cornerstones for large-scale biological computations in living cells.

## Results

**Design rationale for novel intercellular signaling channels**. To design novel intercellular signaling channels in *E. coli*, we took advantage of the enormous pool of secondary metabolites and bacterial aTFs that are responsible for secondary metabolite regulation[32,33]. An initial screening of the Kyoto Encyclopedia of Genes and Genomes (KEGG) database and literature was conducted following these criteria: (i) the signal molecules should be biologically synthesized small molecules that are presumably able to freely diffuse across cell membranes; (ii) the signal molecules should be sensed by aTFs with high specificities; (iii) the precursors of the signal molecules must be universal intracellular metabolites in both prokaryotic and eukaryotic cells; and (iv) the total number of enzymes for synthesizing a signaling molecule is minimal.

The screen yielded ten candidates for intercellular signaling channels. For each candidate, the receiver module consisted of an aTF and the corresponding operator (namely, the promoter segment to which the aTF binds), and the sender module contained genes required for the biosynthesis of the signaling molecule, which were gleaned from diverse species including *Pseudomonas*, *Rhodobacter*, *Streptomyces*, *Photorhabdus*, *Bradyrhizobium*, *Yersinia*, and higher plants. All the candidate signal molecules were supposed to be synthesized from universal cellular metabolites, such as amino acids and central-carbon metabolism molecules, via minimal biosynthetic pathways. For example, salicylate (Sal) as a candidate signaling molecule can be sensed by NahR from *Pseudomonas putida* that utilizes Sal as a carbon source[34]. From the KEGG database, we found two potential Sal biosynthesis pathways using a universal precursor, chorismate in the shikimate pathway: (i) a dual-gene operon *pchBA* from *Pseudomonas aeruginosa*[35] and (ii) a single gene *irp9* from *Yersinia enterocolitica*[36]. Another example is the candidate molecule isovaleryl-HSL (IV) synthesized from a branched-chain amino acid, isoleucine, via a pathway consisting of *bdkFGH*, *IpdA1*, and *bjaI* genes. The first four biosynthetic genes lies in the *IpdA1-bdkFGH* operon from *Streptomyces avermitilis*[37], whereas the last one and the corresponding IV-sensing aTF, *bjaR*, are from *Bradyrhizobium japonicum*. Similarly, for the candidate molecule *p*-coumaroyl-HSL (pC), an incomplete biosynthetic pathway exists in the plant-symbiont bacterium *Rhodobacter*

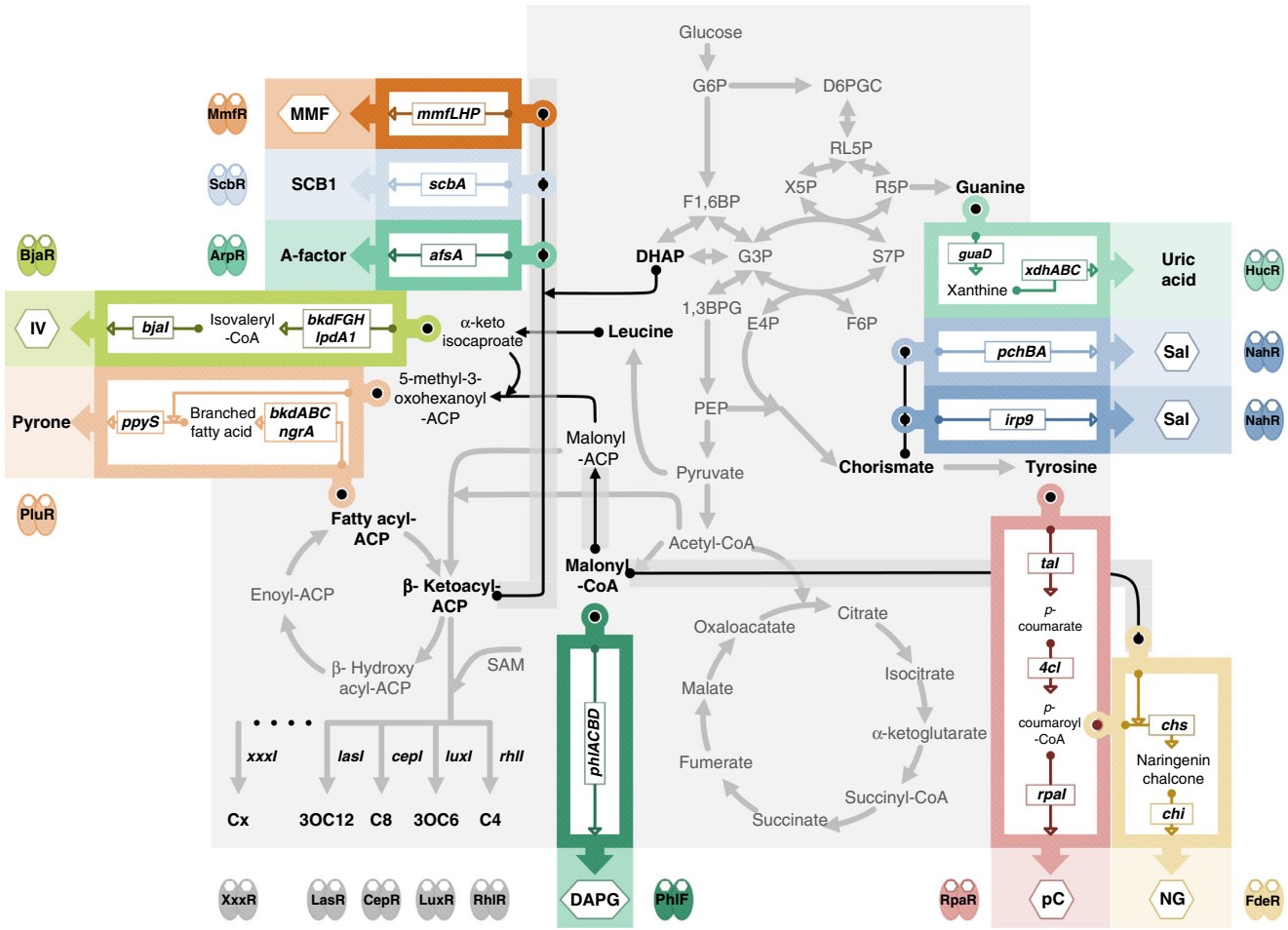

**Fig. 1 Design schemes of gene cassettes for synthesizing signal molecules.** The signal molecules for signal molecules synthesized from central metabolites are shown in shaded hexagons, with their respective receiver aTFs shown in dimerized ovals. The central metabolic pathways are shown in light gray, whereas the designed gene cassettes for each signal molecule are highlighted with different colors. The classic quorum sensing molecules and their biosynthetic pathways are also shown in gray.

palustris[38]. We completed the pathway with the gene *tal* from *Rhodobacter sphaeroides*, which uses a universal amino acid, tyrosine, as the precursor. It is worthy to note that pC as a synthetic signaling molecule has been reported in previous studies[27,39] but its complete biosynthesis, to our knowledge, has not been reported before. To obtain workable promoters for the receiver modules, the operators of aTFs were combined with several variants of the core promoter, from which the optimal combination was selected for each candidate channels (Supplementary Figs. 1 and 2). Based on the same strategy, we implemented the design of the rest candidate channels, using 2,4-diacetylphloroglucinol (DAPG), methylenomycin furan (MMF), naringenin (NG), uric acid, SCB1, A-factor, and pyrone as the signal molecules, respectively (Fig. 1)[40–49]. The design details of all ten candidate channels are listed in Supplementary Table 1. Besides, several well-known natural QS systems were also systematically optimized and added to our intercellular signaling toolbox. These included $C_4$-HSL (C4), 3-oxo-$C_6$-HSL (3OC6), $C_8$-HSL (C8), and 3-oxo-$C_{12}$-HSL (3OC12) systems using RhlR, LuxR, CepR, and LasR as the sensing aTFs, respectively.

**Characterization of the intercellular signaling candidates**. We next set out to quantitatively characterize the candidate channels by evaluating the dynamic range of the signal-sensing promoters. We constructed the *E. coli* sender cell lines with the biosynthesis

gene cassettes of the signal molecules and a constitutively expressed red fluorescent protein (RFP). The receiver *E. coli* cells expressed a green fluorescent protein (GFP) under the control of the responsive promoter as an indication of channel output. The dynamic range of each responsive promoter in the receiver cells was measured by GFP intensity in response to varied concentrations of signal molecules from the sender cells. Two different strategies were adopted to vary the ratio of the sender and the receiver cells[28]: simultaneously adding sender and receiver cells in a co-cultured system with different initial ratios, or adding receiver cells into a fresh medium supplemented with the supernatant of the overnight-cultured sender cells (Supplementary Fig. 5). We found that the two strategies generated similar results (Supplementary Fig. 6). In six of the ten candidate channels, the responsive promoters were significantly activated by the cognate signal molecules generated by the sender cells, reaching dynamic ranges of 1380-fold, 47-fold, 170-fold, 350-fold, 16-fold, and 26-fold for DAPG-, Sal-, pC-, IV-, NG-, and MMF-channels, respectively (Fig. 2a). In particular, the Sal signal were successfully produced by two designed biosynthetic pathways (*irp9* and *pchBA*). Unfortunately, other four channels (uric acid, SCB1, A-factor, and pyrone) failed to achieve cell–cell communication between the sender and receiver cells (Supplementary Table 1).

The four classic QS channels which were added to our toolbox had dynamic ranges from 20- to 40-fold. We created promoter libraries and selectively enhanced their dynamic ranges. For

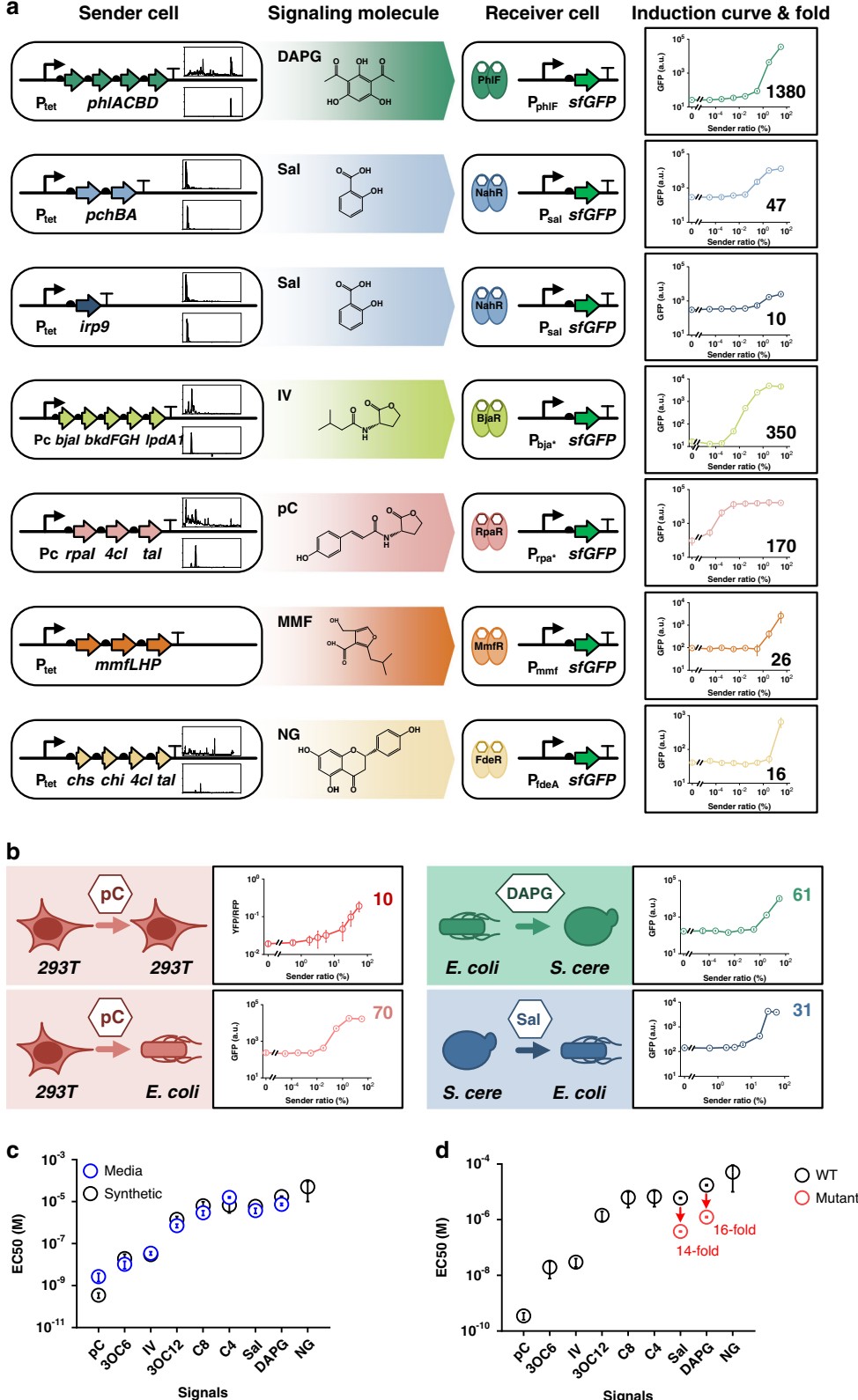

example, for the C8-CepR channel, we re-designed it by combining the CepO operator sequence and a core *lux* promoter in five different versions[50] (Supplementary Fig. 3a). The results showed that the dynamic range of the best responsive promoter increased about twofold in comparison with the wild-type promoter (Supplementary Fig. 3b). The same design strategy was adopted to optimize other three QS channels (Supplementary Table 2). The dynamic ranges of the optimized QS channels increased to 82-, 124-, 150-, and 185-fold for the C4-, 3OC6-, C8-, and 3OC12-channels, respectively (Supplementary Fig. 4). Thus, we successfully acquired six designed and four optimized intercellular signaling channels with wide dynamic ranges.

**Fig. 2 Characterization and optimization of cell–cell communication channels. a** Quantitative characterization of the six de novo designed cell–cell communication channels. The sender cells expressed biosynthesis genes of signal molecules whose HPLC-MS spectra are shown in the upper insets with those of reference chemically pure signal molecules in the lower insets. The receiver cells expressed receptor aTFs and the reporter GFP under the cognate promoters. The last column shows the dose–response curves of the receiver cells in co-culture systems with sender cells introduced at different ratios. The numbers in each sub-figure are maximal fold changes for each dose–response curve. **b** Characterization of cross-species communication via the de novo designed channels. The pC, DAPG, and Sal channels were transferred to yeast and human cells for cross-species communications. For human HEK-293T receiver cells, YFP fluorescence was normalized by the red fluorescence of a constitutively expressed RFP gene on the same plasmid (*293T*, human HEK-293T cells; *S. cere*, *S. cerevisiae*). **c** The sensitivity of each channel as measured by EC50 on the *E. coli* dose–response curves with chemically synthesized signal molecules (black circles) or those quantified in the sender-cell culture media by HPLC (blue circles). **d** The improvement of sensitivities for the DAPG and Sal signaling systems by directed evolution. Red circles indicate the EC50 of the optimized PhlF and NahR mutants for the DAPG and Sal systems, respectively. Both black and red circles were measured with chemically synthesized pure molecules. Data represent the mean fluorescence of three replicates, and error bars show the SD of each measurement. Source data are provided as a Source Data file.

**Quantifying and improving the signaling sensitivity.** The sensitivities of the responsive promoters and aTFs in the receiver cells were defined by the half-maximal effective concentration (EC50) that could be obtained from the dose–response curves of the promoter activities as functions of chemically pure signal molecules (Supplementary Figs. 7 and 8). By fitting the dose–response curve to a simple activation model, the EC50 values of the tested receiver modules were extracted as $4 \times 10^{-10}$, $2 \times 10^{-8}$, $3 \times 10^{-8}$, $1.4 \times 10^{-6}$, $6 \times 10^{-6}$, $6.3 \times 10^{-6}$, $6.7 \times 10^{-6}$, $1.7 \times 10^{-5}$, and $5.1 \times 10^{-5}$ mol/L for the pC-RpaR, 3OC6-LuxR, IV-BjaR, 3OC12-LasR, Sal-NahR, C8-CepR, C4-RhlR, DAPG-PhlF, and NG-FdeR channels, respectively (Fig. 2c). Unfortunately, we failed to chemically synthesize sufficient MMF molecules for this assay. Among the nine channels tested, the pC-RpaR, 3OC6-LuxR, IV-BjaR, and 3OC12-LasR channels were the best candidates for intercellular communication, since their higher sensitivities require small amounts of signals to be synthesized by the sender cells and thus impose lower metabolic burdens. We also quantified the signal molecules synthesized by the sender cells by high-performance liquid chromatography-mass spectrometry (HPLC-MS) and determined the EC50 values from the dose–response curves in co-culture experiments (Supplementary Fig. 9). We found they were consistent with those obtained by chemically pure signal molecules, indicating the communications were not activated by spurious substances (e.g., metabolites or cellular discharges) in the cell culture and thus were highly specific (Fig. 2c).

Subsequently, we tried to improve the sensitivity of signaling channels DAPG-PhlF and Sal-NahR, of which the EC50 values were on the order of micro-molar and above through directed evolution. The signal-sensing aTFs, *phlF* and *nahR* for the DAPG-PhlF and Sal-NahR channels, respectively, were randomly mutated and subject to a dual selection regime with or without the signal molecules, based on the fluorescence of the responsive promoter-reporter (P$_{phlF}$–*sfgfp* and P$_{sal}$–*sfgfp*) cassettes by flow cytometer (Supplementary Fig. 14). After three to six rounds of selection, several PhlF and NahR mutants were picked up and measured for their EC50 values for the cognate signal molecules (Supplementary Fig. 14). We found that a D128G mutation consistently existed in 5 improved *phlF* mutants, which decreased the EC50 value by more than 10-fold (Fig. 2d), whereas a Q168R mutation was essential for all improved NahR mutants with their EC50 values decreased by about 15-fold (Fig. 2d). These results suggested that the D128G and Q168R mutations might be key to improving the sensitivity of PhlF and NahR to the DAPG and Sal molecules, respectively, highlighting the role of single amino acid mutations in the improvement of sensitivities up to tenfold for signaling system design (Supplementary Fig. 15).

On the other hand, by optimizing the biosynthetic cassettes in the sender cells, we increased signaling molecule production by 10- to 1000-fold for the best combinations of promoters and

ribosome-binding sites (Supplementary Fig. 10). Overall, these optimizations enhanced the robustness of channel activation.

**Orthogonality of the ten intercellular signaling channels.** When multiple communication channels are integrated in the same genetic circuit, interference might occur at the signal-sensing and the promoter-responding levels. Conventionally, they have been defined as signal cross-talk and promoter cross-talk, respectively[27]. We experimentally characterized the signal and the promoter cross-talks for all the above ten signaling systems (Fig. 3a).

To examine the cross-talk at the signal level, 100 sender–receiver pairs were measured for their fold change in co-culture systems. The fold change of each sender–receiver pair was defined as the ratio of the induced and non-induced expression of the responsive promoters (see the "Methods" section for more details). As shown in Fig. 3b, the four well-studied QS channels (C4, 3OC6, C8, and 3OC12, highlighted in the yellow box) exhibited extensive cross-talks (as high as 49.7-fold with an average of 18.5-fold mis-induction), but the six de novo designed channels (highlighted in the red box) had significantly lower cross-talks (as high as 5.4-fold with an average of 1.67-fold mis-induction). The remarkable orthogonality is presumably due to the highly diverse chemical structures of these signal molecules. It was worthy to note that the 3OC12 channel was orthogonal with the C4 and 3OC6 channels, but not orthogonal with the C8 channel (Fig. 3b and Supplementary Table 6). After removing the C4 and C8 channels from the toolbox, the remaining eight channels became orthogonal with each other at the signal level (Fig. 3c). To visualize orthogonality, we placed each sender colony on an agar plate surrounded by all ten receiver colonies (Fig. 3d). By imaging with fluorescent stereoscopy, we observed that all ten sender colonies could activate the reporter gene expression in their cognate receiver cells, with only occasionally weak activation observed in non-cognate receiver cells. The colony activation patterns were consistent with the results in liquid medium (Fig. 3b, d).

To characterize the cross-talk at the promoter level, we co-transformed pairs of aTFs and their responsive promoter-reporter cassettes into *E. coli*, resulting in 100 hybrid receiver strains (Fig. 3e). For each hybrid strain, an aTF was induced by the cognate signaling molecule in an exo-supplemented manner. Similar to the signal cross-talk results, the de novo-designed channels exhibited minimal promoter cross-talks (as high as 2-fold with an average of 1.07-fold mis-induction, as highlighted in red box in Fig. 3f), whereas aTFs for the traditional straight-chain AHL channels interfered extensively with each other (as high as 377-fold with an average of 67.3-fold mis-induction, as highlighted in yellow box in Fig. 3f) (Supplementary Table 7). After removing the C4 and C8 channels, we found that the aTFs

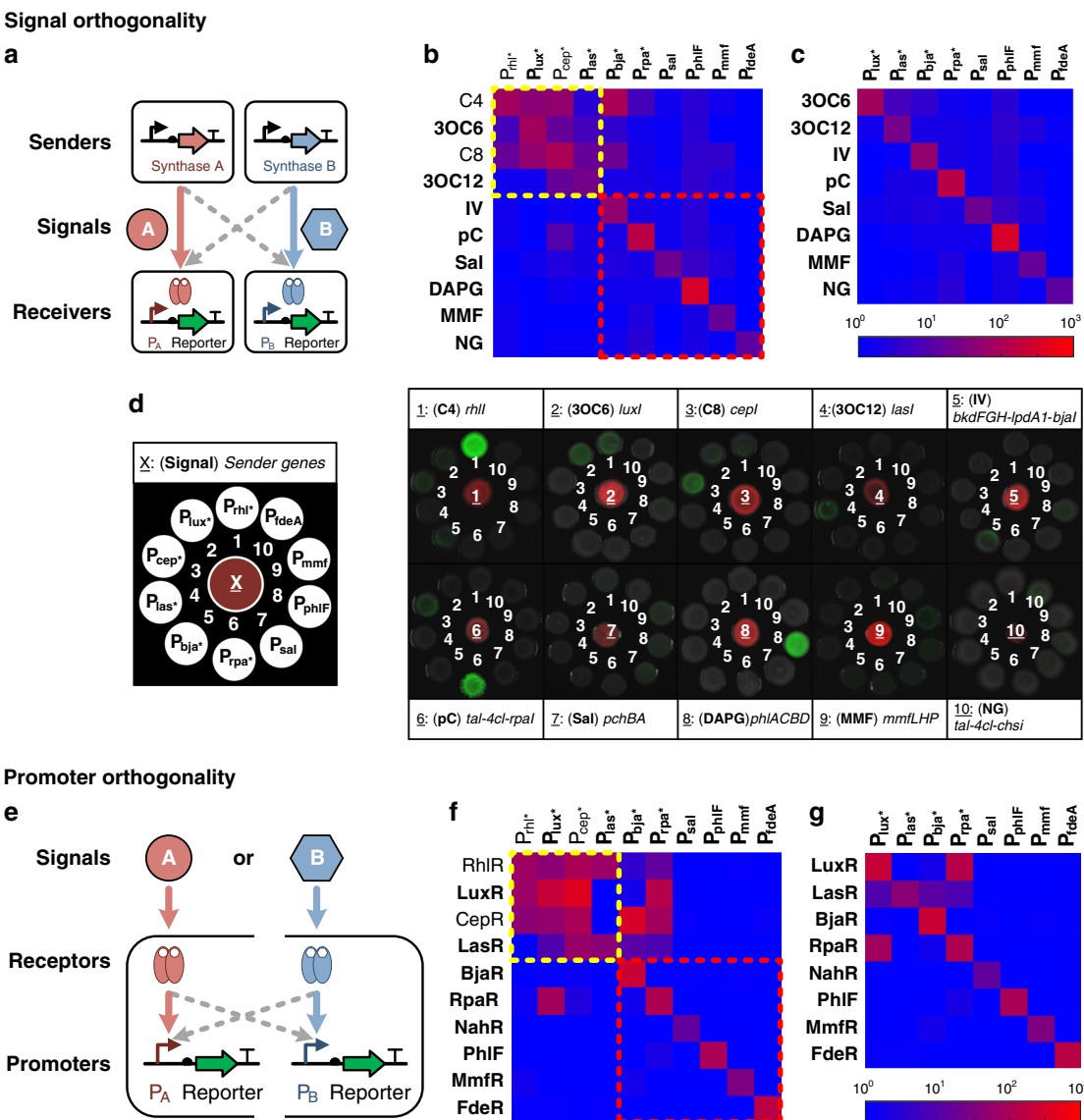

**Fig. 3 Signal and promoter orthogonality among cell–cell communication channels. a**, **e** Diagrams of orthogonality characterization of signal molecules (**a**) and responsive promoters (**e**). Gray arrows indicate possible cross-talks between two channels. **b** Characterization of signal orthogonality among all ten channels. The fluorescence of each receiver cell was measured by flow cytometry. Heatmaps represent the fold change of fluorescence before and after induction. **c** Eight channels were selected to comprise a set of orthogonal communication systems. **d** Signal orthogonality measured on agar plates. Each sender cell strain (constitutively expressing RFP) was inoculated on the center of an agar plate, surrounded by all ten receiver cell strains (expressing GFP from cognate or non-cognate responsive promoters). Channels are labeled from 1 to 10, with the underlined numbers indicating the channel number in the sender cells. Fluorescent and bright-field images were merged. **f** Characterization of promoter orthogonality among all ten channels. **g** The same eight channels as in **c** are shown for promoter orthogonality. All data were acquired from at least three independent replicates. Source data are provided as a Source Data file.

sensing RpaR and LuxR could mutually activate each other's promoter (Fig. 3g). Thus, the RpaR-P$_{rpa*}$ and LuxR-P$_{lux*}$ systems should not coexist in the same cell for engineering purposes.

**Cross-kingdom capability of the designed channels**. To achieve cross-species and cross-kingdom communication, both sender and receiver modules should readily function in various cell models. As proofs of concept, we transferred some of the de novo designed communication channels from *E. coli* to other prokaryotic and eukaryotic cells, especially to human cell lines for applications such as artificial tissues and smart therapeutic cells. As the most sensitive signaling channel in our toolbox, the pC-

RpaR channel was chosen to be transferred into HEK-293T cells, a robust cell line derived from human embryonic kidney cells. The receiver module in HEK-293T cells was constructed by fusing a VTR3 activation domain[51] to RpaR and creating an RpaO-CMV promoter with two RpaR-binding sites placed upstream of the minimal CMV promoter. We found that the pC signal synthesized from sender HEK-293T cells activated the modified RpaO-CMV promoter in the receiver HEK-293T cells by about tenfold (Fig. 2b and Supplementary Figs. 11 and 12). By replacing the *E. coli* promoter with species-specific ones, we established cross-kingdom communications including (i) the DAPG-PhlF channel from *E. coli* to *S. cerevisiae*, (ii) the Sal-NahR channel from *S. cerevisiae* to *E. coli*, and (iii) the pC-RpaR

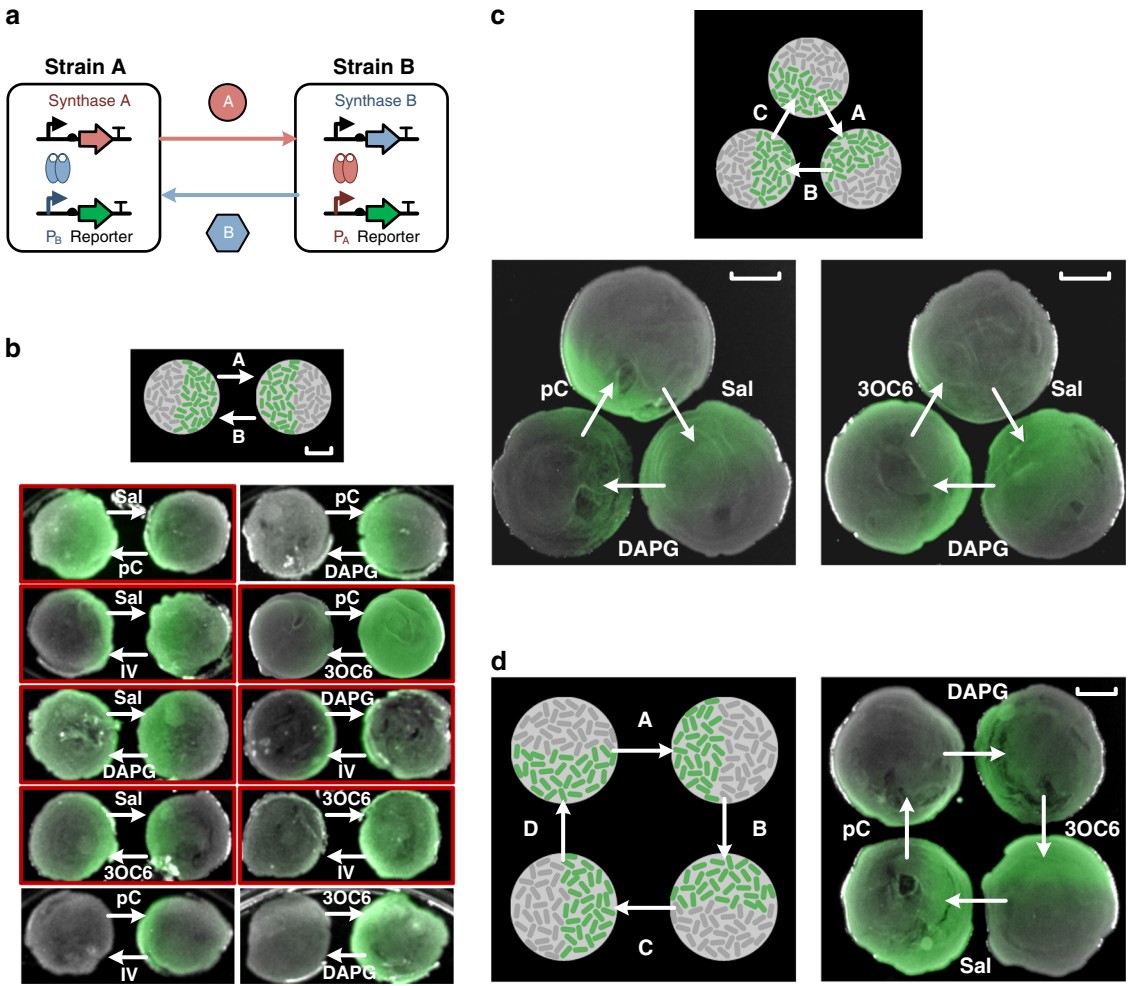

**Fig. 4 Simple multi-channel communication circuits to form spatial patterns. a** Diagram of two mutually communicating *E. coli* strains with orthogonal channels. The diagrams for three- and four-channel circuits are shown in Supplementary Fig. 17. Each strain contained the sender module of one channel and the receiver module of another. **b** Two-channel communication circuits for all ten combinations of the five channels. Fluorescent and bright-field pictures of the colonies were merged. Colonies that formed the correct patterns are highlighted by red boxes. **c, d** Three-channel (**c**) and four-channel (**d**) communication circuits with sender and receiver modules of each channel expressed in neighboring populations in a clockwise direction. Two and one colony patterns are shown for three- and four-channel circuits, respectively. The scale bars correspond to 0.5 cm.

channel from HEK-293T to *E. coli*. All communication channels were successfully activated between the heterotypic sender and receiver cells (Fig. 2b and Supplementary Fig. 13). Together, we have demonstrated the universality of these channels and their potential for cross-kingdom communication.

**Spatial pattern of multi-channel intercellular communication.** With the cell–cell communication toolbox, we managed to build simple genetic circuits that involved two-, three-, and four-channel communications to demonstrate their modularity. First, we designed two-channel communication circuits in pairs of different strains, each containing the sender module of one signal and the receiver module of the other (Fig. 4a). Thus, paired strains could be mutually activated along the gradient of signals synthesized from the other end (Fig. 4b). Experimentally, we inoculated each pair of strains in spatially separated 1.5cm-diameter spots on an agar plate. Seven two-channel circuits successfully formed face-to-face gradient fluorescence patterns (Fig. 4b). We further designed three- and four- channel circuits, in which the inter-communicating strains were arranged in sender–receiver loops in triangular and square-shaped patterns (Fig. 4c, d and Supplementary Figs. 16 and 17). In these

experiments, each colony received the strongest signals from both of its nearest neighbors, but responded to only one signal, producing a biased/asymmetric fluorescence pattern. Among all tested designs, two three-channel circuits (pC-Sal-DAPG and 3OC6-Sal-DAPG) and one four-channel circuit exhibited the expected triangle-looped and square-looped activation patterns, respectively (Fig. 4c, d).

**Complex logic gate circuits built with multiplexed signals.** To demonstrate more complex multicellular biocomputing functions, we took a sophisticated three-input XOR-AND logic-gate circuit as an example. As shown in Fig. 5a, the XOR-AND logic-gate circuit was deployed in seven different *E. coli* strains coordinated by four communication channels. Each strain contained a NOR gate (cell-1 to cell-6) or a Buffer gate (cell-7) in a spatially distributed manner[52]. The first three NOR gates (cell-1, cell-2, and cell-3) combined into an AND-gate circuit (Supplementary Fig. 18a) and the next three NOR gates (cell-4, cell-5, and cell-6) and the last Buffer gate (cell-7) made up the XOR-gate circuit[52] (Supplementary Figure 18d). Each NOR gate was configured with two input promoters that responded the upstream signals and one output biosynthesis gene cassette to generate the signal

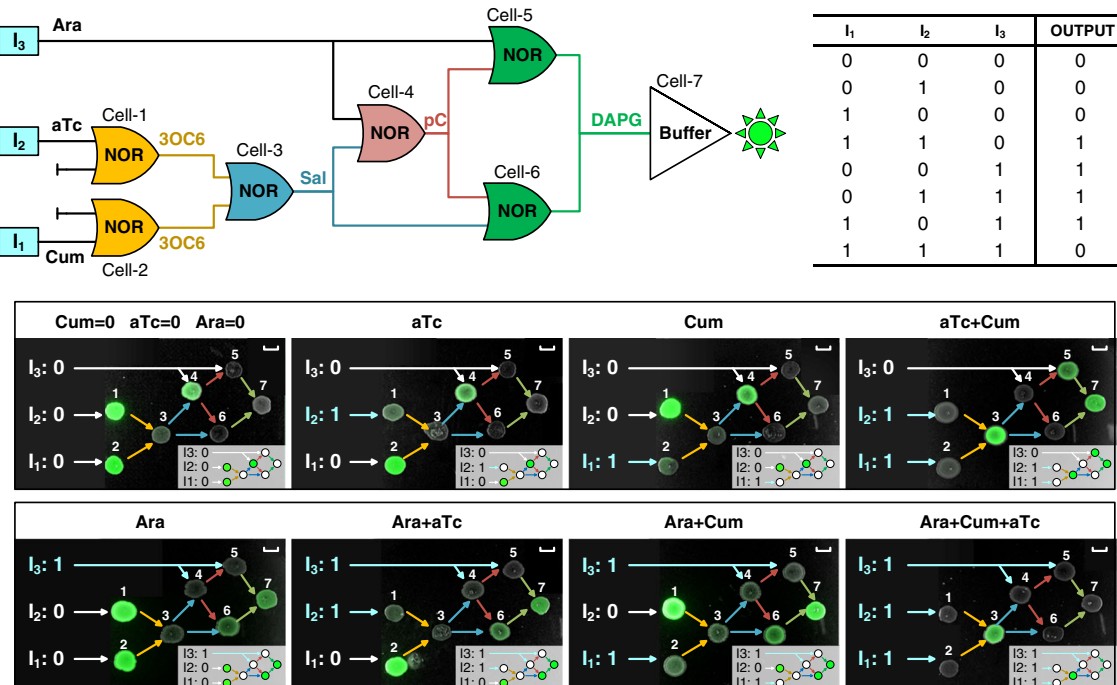

**Fig. 5 A distributed biocomputing circuit built with multiplexed signals. a, b** The scheme (**a**) and truth table (**b**) of the engineered three-input AND–XOR logic gate circuit built with seven genetically different strains and four orthogonal channels. **c** Experimental and theoretical (insets) logic outputs of the final and all intermediate computation results as shown by spatially distributed colonies on the agar plate. The scale bars correspond to 0.5 cm.

molecules. In addition, all strains had a built-in core regulatory component–the lambda repressor gene and its cognate responsive promoter which drove a *gfp* gene to report for intermediate computation results[52]. We first independently constructed all the strains, and then spotted them on agar plates with the spatial arrangements required to perform the sequential XOR- and AND-gate functions, respectively. Both circuits successfully performed their logical operations for all possible combinations of the two inputs (Supplementary Figs. 18c and 17f). The XOR gate and AND gate were directly connected on agar plate to form a three-input AND–XOR logic gate circuit. We found that not only the final outputs of the AND–XOR gate agreed with expectation in the truth table (Fig. 5b, c), such spatial layout allowed for the visualization of intermediate computation results, which also turned out to be correct (inserted schematics of each sub-figure in Fig. 5c and Supplementary Figs. 19 and 20). To our knowledge, this is the first engineered biocomputing circuit that simultaneously utilizes four communication channels.

## Discussion

Cell–cell communications are ubiquitous in nature[2,53,54]. From an engineering perspective, these widespread communication systems provide a vast reserve of potential synthetic communication parts including signal molecules, highly specific receptors and aTFs[55]. However, naturally evolved parts are not perfect for synthetic gene circuit construction. In this study, we proposed a de novo design route for synthetic intercellular communication channels. By rational design and directed evolution approaches, we established a toolbox of biochemical channels that can be used for multi-channel communications in applications involving pattern formation and distributed cellular bio-computation.

Most natural intercellular communication systems are species- or kingdom-specific[23]. For example, previous efforts were made to transfer the bacterial AHL systems into mammalian cells in order to acquire orthogonal intercellular signaling systems for artificial tissue and organ engineering. Unfortunately, the essential

precursors (acyl-ACP) in mammalian cells are locked in by the Type II fatty acid synthesis multi-domain enzymes and not available for the biosynthesis of the AHL molecules[56]. In our toolbox, however, the pC and IV molecules were synthesized from canonical amino acid (i.e., L-tyrosine and L-leucine) sources despite being structurally similar to AHLs. The successful transfer of the pC channel to human HEK-293T cells highlights the design rationale of diverting common cellular metabolic pathways for synthetic circuits, and we expect the de novo designed channels, including IV, DAPG, MMF, and NG, to apply to mammalian systems as well.

We also optimized the receiver modules by directed evolution for better dynamic ranges and sensitivities, to reduce the metabolic burden to the sender cells (Supplementary Figs. 22 and 23), as well as to improve channel compatibility for eukaryotic receiver cells. These dedicated channels with microbial and plant origins would be especially suited for mammalian systems, because they would not interfere with endogenous signaling systems as those based on dopamine and histamine[12] would. On the other hand, although recent work indicates that natural QS signaling in bacterial pathogens is tap-wired by the host AhR signaling pathway in various vertebrates for immunomodulation[57], our designer signal molecules may not cause unwanted host responses because they are structurally different from natural QS molecules and may thus evade host surveillance.

Inspired by electric circuits and telecommunications, where channels are spatially insulated or functions in different wavebands, we took advantage of the enormous chemical space of biologically derived molecules for channel insulation, which was further enhanced by optimizing the specificity of the receiver modules[5,23]. The success of our construction underscores a general principle that naturally occurring biochemical machineries have merely exploited all possible solutions, leaving almost boundless design space for synthetic biological construction. A recent study on engineered kinases has also supported the view[31].

Universality and orthogonality together constitute the essence of modular design for complex synthetic biological functions. In

two examples, we showed the use of these modular channels to spatially and logically organize different computing units. We successfully implemented up to four channels in an engineered cell consortium, which to our knowledge is the largest in multi-cellular computing studies[28,52]. Notably, in the second example, channels were serially connected to form computing cascades. Currently, there are maximally two-channel modules implanted in a single cell. The fact that our individual modules imposed minimal metabolic burden to the host cell could enable engineered communication hubs with possibly more than two channels intersecting in a single cell. The exact limit of this number remains an open question, but it ultimately defines the information processing complexity of cell consortium computation and its real-world application potential.

Intercellular communication plays a pivotal role in expanding the engineered functions from single cellular behaviors to multicellular artificial tissues, microbiome therapy such as in the human gastrointestinal tract or in tumors[58] and general biocomputing systems[22,25,59,60]. Our study has demonstrated the possibility of engineering natural secondary metabolites and signaling modules into dedicated intercellular communication channels. With an expanded toolbox of modular channels, more sophisticated circuits could be designed in mammalian cell lines to implement stable multi-input, multi-output, and structurally organized computing systems for in vivo therapeutic applications.

## Methods

**Microbial genes, strains, and culture conditions.** All the heterologously expressed genes in *E. coli* and *S. cerevisiae* were synthesized or amplified from the genome of target species using the primers listed in Supplementary Table 4 and 5. All the sender and receiver cassettes (Promoter-Gene-Terminator) used in *E. coli* and *S. cerevisiae* are listed in Supplementary Data 1. *E. coli* strain DH5α (*fhuA2 lac (del)U169 phoA glnV44 Φ80′ lacZ(del)M15 gyrA96 recA1 relA1 endA1 thi-1 hsdR17*) was cultured in LB Broth or M9 minimal medium (6.8 g/L Na$_2$HPO$_4$, 3 g/L KH$_2$PO$_4$, 0.5 g/L NaCl, 1 g/L NH$_4$Cl, 2 mM MgSO$_4$, 100 µM CaCl$_2$, 0.4% glucose, 0.2% casamino acids, and 340 mg/L vitamin B1) in 96-well shaking incubators (1000 r.p.m) at 37 °C. *S. cerevisiae* strain WP125 (*W303 MATa rtTA far1Δ his3 trp1 leu2 ura3*) was cultured in YPD medium (1% yeast extract, 2% peptone, 2% glucose) in 500 ml flasks (shaking at 225 r.p.m.) at 30 °C. All media and agar plates contained 100 µg/ml Ampicillin, 50 µg/ml kanamycin and/or 25 µg/ml chloramphenicol unless stated otherwise. Isopropyl β-D-1-thiogalactopyranoside (IPTG) (0.1 mM or 1 mM) was supplemented to culture media (Supplementary Fig. 7). For strains containing P$_{tet}$ promoter, 200 ng/ml anhydrotetracycline (aTc) was added to culture media.

**Mammalian genes, cell culture, and transfection.** The *tal*, *4cl*, and *rpaI* genes were directly cloned from the plasmid used in *E. coli*. The *VTR3* gene was a kind gift from Professor Zhen Xie's Lab in Tsinghua University, Beijing, China. The lentiviral vectors (pspAX2, pCMV-dR8.91 and ML280) and reporter genes (*iRFP, mTurquoise2, Citrine, mCherry*) were gifts from Professor Yihan Lin's Lab in Peking University, Beijing, China. All the sender and receiver cassettes used in HEK-293T cells are listed in Supplementary Data 1, constructed with the primers listed in Supplementary Tables 4 and 5. Human embryonic kidney cells (HEK-293T, ATCC: CRL-11268) used for mammalian cell–cell communication were cultured in high-glucose Dulbecco's modified Eagle's medium (DMEM, Gibco) complete media containing 4.5 g/L glucose, 10% FBS (Life Technologies), 0.045 unit/ml penicillin, and 0.045 unit/ml streptomycin at 37 °C, 100% humidity and 5% CO$_2$.

For transient transfection, cells were transfected using optimized polyethyleneimine (PEI "Max", 1 mg/ml in water; Polysciences, Eppelheim, Germany). One day before transfection, ~1.5 × 10$^5$ HEK-293T cells were seeded into 12-well plates. The transfection mixture was prepared by adding 1.6 µg plasmid and 4.8 µl PEI reagent into 200 µl serum-free DMEM for each well. The transfection mixture was added dropwise after incubation for 10 min. Three hours after transfection, the culture medium in each well was replaced with fresh DMEM complete media. Cells were then cultured for an additional 48 h before harvesting.

To generate stable sender cell lines, HEK-293T cells were first transfected with one of the two lentiviral plasmids containing synthesis genes and reporter genes using the transient transfection protocol described above. After 48 h, 1 ml culture medium was harvested from each well. For lentivirus infection, both lentiviruses (each in 1 ml culture media) were added together into ~8.5 × 10$^5$ cells (2 ml culture media) seeded in a 6-well plate one day before infection. After 48 h, cells were harvested for flow cytometric sorting. Flow cytometric sorting was performed using a BD FACSAria IIIu (BD Biosciences, Franklin Lakes, NJ). The iRFP and mTourquoise2 reporter fluorescence were detected with the Alexa Fluor 700 and BV421 channels, respectively. The double-positive population was sorted as stable sender cell lines.

**Synthetic signaling molecule induction.** For microbial cells, receiver cells were first diluted 200-fold from overnight cultures in LB medium (*E. coli*; 16 h) or YPD media (*S. cerevisiae*; 24 h), and then cultured for an additional 3 h. After 3 h, receiver cells were diluted 200-fold into M9 (*E. coli*) or YPD medium (*S. cerevisiae*) supplemented with appropriate concentrations of corresponding synthetic signaling molecule, and then cultured for an additional 12 h (*E. coli*) or 24 h (*S. cerevisiae*) before being measured by flow cytometry. For human HEK-293T cells, cells were first transfected with receiver plasmids as described above, and then appropriate concentrations of the inducer was added into wells as supplement of the fresh complete medium 3 h after transfection. Cells were then cultured for an additional 48 h before being harvested for flow cytometric analysis. IV-HSL was chemically synthesized (Supplementary Fig. 21). All other synthetic signaling molecules used were purchased from Sigma-Aldrich (St. Louis, Missouri, USA).

**Sender media induction and co-culture.** Supernatant of the sender culture media was prepared using the following methods. *E. coli* senders were cultured overnight for 12 h in M9 medium supplemented with 200 ng/ml aTc where applicable. *S. cerevisiae* senders were cultured for 96 h in 500 ml flasks containing 50 ml YPD supplemented with 1 mg/ml aTc. 5 ml 20% glucose was supplemented to each flask at 24, 48, and 72 h. Human HEK-293T sender cells were cultured in 10 cm dish for 48 h in complete DMEM media supplemented with 0.5 g/L tyrosine. By the end of each culture, cell-free culture media was obtained, by centrifuging if necessary. Supernatant of all species was filtered with 0.2 µm sterile filter. Each culture medium was diluted by equal volume of fresh M9 medium (*E. coli*), YPD medium (*S. cerevisiae*), or complete DMEM (human HEK-293T). The diluted supernatant was further subject to twofold or tenfold serial dilution using the same medium before mixing with the corresponding receiver cells. For HEK-293T cells, the diluted supernatant was added into wells 3 h after transfection, replacing the culture media with transfection reagent. For all other species, receiver cells were diluted 200-fold into the corresponding supernatant. After mixing with the sender supernatant, the receiver cells were cultured for an additional 12 h (*E. coli*), 24 h (*S. cerevisiae*), or 48 h (human HEK-293T) before being measured by flow cytometry.

In co-culture systems, *E. coli* sender and receiver cells were first diluted 200-fold separately into M9 medium from overnight culture in LB medium, and then cultured for 3 h. Subsequently, sender cells were diluted by control cells (With RFP reporter, without sender genes) with a twofold or tenfold serial dilution, before mixing with an equal volume of receiver cells. The sender–receiver mix was diluted 200-fold into fresh M9 medium with appropriate concentrations of inducers (IPTG, aTc) and antibiotics. Cells were collected for flow cytometry after being cultured for an additional 12 h.

**Flow cytometric measurement and data analysis.** The fluorescence of all samples was measured by BD LSRII flow cytometer (BD Biosciences, Franklin Lakes, NJ) equipped with high-throughput screening instrument. Data of *E. coli* and *S. cerevisiae* samples were analyzed using FlowJo (TreeStar, Inc., Ashland, OR). Sender (RFP+) and receiver (RFP−) cells were distinguished based on their intensity of red fluorescence. The output value of each sample was defined as the mean fluorescein isothiocyanate value of the RFP− population. Alternatively, flow cytometry data of HEK-293T cells were analyzed using Matlab 2018b (The MathWorks, Inc., Natick, MA). With RFP being the baseline, the output value of each sample was defined as the ratio between its yellow fluorescence (Citrine) and red fluorescence. The mean autofluorescence of a non-fluorescent cell control was also subtracted from the mean value of each sample. All data represent the mean fluorescence of at least three replicates and error bars correspond to the SD of each measurement.

**HPLC-MS quantification.** To prepare the supernatant of the sender culture media, *E. coli* sender cells were first cultured overnight in 5 ml M9 medium and then centrifuged (3000 r.p.m., 2 min), filtered with a 0.2 µm sterile filter, freeze-dried with a Biocool FD-1D-80 vacuum freeze dryer (Biocool, Beijing, China) until completely dried, and finally reconstituted with 500 µl methanol. HPLC-MS analysis was performed with an Agilent Eclipse plus C18 reverse-phase column (2.1 × 100 mM, 3.5 µm) instrument (Agilent Technologies, Santa Clara, CA) for Sal or a Phenomenex Synerg Hydro-RP 80 A LC column (2 × 150 mM, 4 µm) (Phenomenex, Torrance, CA) for the rest signal molecules. Each column was connected to an Agilent 1200 HPLC instrument (Agilent Technologies, Santa Clara, CA). HPLC separation of all molecules was performed from a 5 µl sample under the gradient elution mode with a flow rate of 0.4 ml/min at 25 °C. The mobile phase A and B were 8/92 acetic acid/water and acetonitrile, respectively. Gradient elution was conducted under the following conditions: 25% B for 5 min, 25–100% B with a linear gradient for 5 min, 100% B for 3 min, 100–25% B with a linear gradient for 1 min, then 25% B for 2 min. The outflow was routed to an AB SCIEX Qtrap 4500 mass spectrometer (AB Sciex LLC, Ontario, Canada) equipped with an electrospray ionization (ESI) source and multiple reaction monitoring (MRM). The ESI source operated at negative or positive mode for different molecules (Supplementary Fig. 9). The common mass spectrometric parameters for HPLC-MS/MRM are shown in Supplementary Table 3. Standard curves for each signaling molecule were created with HPLC-MS/MRM by measuring the responses to synthetic signal molecules diluted to a series of known concentrations. The yield of each signaling

molecule from its sender ($c_{max}$) was calculated by fitting with the corresponding standard curve.

**Directed evolution of regulator protein**. First, a mutant library was created by error-prone PCR of the target regulator proteins[61]. The DNA fragments encoding mutated genes were then inserted into a plasmid to be expressed with a J23111 promoter. The plasmid also contained a *sfGFP* reporter controlled by the promoter that binds to the cognate aTF. The mutant library constructs were transformed into *E. coli* and spread on LB plate. For negative selection, the entire library was collected from plate, inoculated into LB medium and cultured for 16 h, and then diluted into M9 medium and cultured for 4~6 h. Low-fluorescent clones were sorted from M9 cultured cells with flow cytometer and spread on LB plate again. For positive selection, the sorted low-fluorescent library was collected from plate and cultured in LB medium for 16 h, and then diluted into M9 medium supplemented with the inducer. Compared to the wild-type clone (with the wild-type regulator), mutant clones which showed stronger fluorescence were sorted by flow cytometry and spread on LB plate. This process of collecting clones from plate, culturing and flow cytometric sorting was repeated for 3 or more rounds. In each round, the inducer concentration for the mutant library decreased by 50% compared to the previous round but remained constant for the wild-type clone. Thus, the inducer concentration for the mutant library was gradually reduced to 50%, 25%, and 12.5% of the initial concentration. In each round of sorting, the mutant clones which had similar or stronger fluorescence compared to the wild-type clone were sorted and spread on plate before entering the next round. To determine the sensitivity of mutant clones, individual clones were picked from each library after positive selections, followed by flow cytometric measurements of induction curves with the method described above.

**On-plate visualization of the colonies**. Agar plates were made with hybrid medium by supplementing LB medium with all the compounds of M9 medium described above. Each strain was cultured overnight in LB medium for 16 h and then diluted 500-fold (orthogonality test) or 10-fold (simple circuits) using LB medium. The dilution factors for some strains were adjusted according to their diverse rates in growth and signaling molecule synthesis. Subsequently, 1 μl (orthogonality test) or 30 μl (simple circuits) of each strain was spotted on agar plate supplemented with appropriate concentrations of IPTG and aTc (Supplementary Fig. 7), forming spots of approximately 0.5 cm (orthogonality test) or 1.5 cm (simple circuits) in diameter. All plates were cultured at 37 °C for 24 h before the fluorescent and bright-field images of each agar plate were captured by homemade multi-color fluorescent imager. All related images were bight-field images merged with the green fluorescent or red fluorescent images.

**On-plate demonstration of AND–XOR logic gates**. Agar plates were prepared using the same hybrid medium (LB + M9) described above. The three inducers (10 mM Ara, 100 ng/ml aTc, 0.1 M cumate) were supplemented into agar plate according to the input signal of each circuit (Supplementary Figure 19). Each strain was cultured overnight in LB medium for 16 h, diluted 100-fold and cultured for an additional 6~8 h until the O.D.600 reached 0.6~0.8. The dilution factors for some strains were adjusted according to their diverse rates in growth and signaling molecule synthesis. Then, 3 μl of each strain was spotted on agar plate, forming a spot of ~0.5 cm in diameter. Spots were positioned in the shapes of regular triangles or squares with the distance between each two spots set to 10 mm. Spotting was done in a step-by-step manner. After the last strain was spotted, all plates were cultured in 37 °C for an additional 12 h before the fluorescent and bright-field images of each agar plate were captured by homemade multi-color fluorescent imager. All related images were bight-field images merged with the green fluorescent ones. Finally, all strains were collected from plate using pipette tips and diluted into 1 ml phosphate-buffered saline (with 2 mg/ml kanamycin) solution for flow cytometric analysis.

**Cell burden analysis**. Cells were first diluted 200-fold into M9 medium from overnight cultures in LB medium (16 h), and then cultured for an additional 3 h. After 3 h, each strain was diluted 200-fold into M9 medium supplemented with the inducer (IPTG or aTc) and then cultured for an additional 24 h. OD600 values were measured every 10 min by a BMG CLARIOstar® Plus microplate reader (BMG Labtech, Ortenberg, Germany) during the final 24 h culture. The IPTG concentrations for receivers were chosen according to the protocol for Synthetic signaling molecule induction (Supplementary Fig. 7).

**Reporting summary**. Further information on research design is available in the Nature Research Reporting Summary linked to this article.

## Data availability

All relevant data supporting the findings of this study are available within the article and its Supplementary Information files or from the corresponding authors upon request. Source data are provided with this paper.

## Code availability

Matlab code for analyzing mammalian flow cytometry data is available at git@github.com:xjpatriot87/Mammalian-data-analysis-with-matlab.git

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

## Acknowledgements

We thank Professor Zhen Xie from Tsinghua University, Professor Yihan Lin from Peking University, and Jingwen Zhou from Jiangnan University for their gifted plasmids. This work was supported by the National Key Research and Development Program of China [numbers 2018YFA0900701 and 2015CB910300], the Key Research Program of the Chinese Academy of Sciences [numbers QYZDB-SSW-SMC050 and KFZD-SW-216], the Strategic Priority Research Program of the Chinese Academy of Sciences [number XDB29040000], Shenzhen Institute of Synthetic Biology Scientific Research Program [DWKF20190009], Shenzhen Key Laboratory of Synthetic Genomics (ZDSYS201802061806209), Guangdong Provincial Key Laboratory of Synthetic Genomics (2019B030301006) and the Natural Science Foundation of China [numbers 31722002, 31700084, 31901063, and 31470818].

## Author contributions

C.B.L., Y.T., and H.Q.Z. supervised the project. P.D., H.W.Z., H.Q.Z., J.Y.H., X.D.L., X.X.L., R.S.W., Y.T., Y.H.X., M.W., L.Q., and Y.H.C. designed and performed the experiments. P.D., L.Q., H.Q.Z., and C.B.L. wrote or revised the manuscript.

## Competing interests

The authors declare no competing interests.
