## [Peer Review File · Nature Communications]

Reviewers' Comments:

Reviewer #1:

Remarks to the Author:

Factual summary

Six novel orthogonal cell-cell signaling channels were ported to *E. coli* from diverse organisms such as *Pseudomonas sp.*, *Yersinia sp.*, *Streptomyces sp.*, *Rhodobacter sp.* and others. Synthesis pathways for the molecules and genetic sensor components were developed and characterized in *E. coli* for use in synthetic biology. The sensor components of two signaling modules were further optimized through directed evolution. The signaling modules were then successfully ported to mammalian HEK-293T cells and yeast *S. cerevisiae* and used for intra- and inter-kingdom communication. Finally, the communications modules were used to construct simple devices in *E. coli*, including a spatial pattern formation system and a spatially distributed biological logic gate system.

Strengths and weaknesses

Some of the much-needed cell-to-cell communication tools for use in Synthetic Biology are developed in this study. Not only are these tools implemented for use in *E. coli*; some are shown to be functional in yeast and mammalian cells and could, thus, be used for inter-kingdom communication. The study also implements original and exciting spatially segregated logic gate circuitry that has the potential to be of good interest to the Synthetic Biology community.

The developed signaling tools have not been assessed for their effects on *E. coli* growth and a metabolic burden analysis is absent. This should be included in the final manuscript. A further weakness is linked to the number of communication channels implemented in organisms other than *E. coli*: only a single communication channel has been implemented in mammalian cells, whereas two have been ported to *S. cerevisiae*. This raises some doubt as to the potential for the portability of the other channels. Finally, the sequences of the developed components are not clearly presented in the Supplementary information; a comprehensive list of all used and developed genetic tools has to be included. Important: the genetic sequences have to be in copiable text format.

Verdict

I endorse the publication of this paper after the following revisions have been implemented:

1. Please provide a complete and comprehensive cell burden analysis; how constructs impact growth in *E. coli* (i.e. provide growth curves).
2. Detailed presentation of all genetic sequences and constructs.
3. Minor, but potentially useful: include characterization of signaling modules in different growth media in *E. coli* (M9, LB, 2xYT).
4. Expand referencing in Discussion.
5. Include plots of actual values presented on the heatmaps.
6. Native English copy-editing

Comments

- Minor language corrections throughout; only a small subset of what has to be done is mentioned here. In general it is OK, but lacks the native English form.
 - Figure 1 legend sounds strange in past tense.
 - pg7, lines 244-onwards; a little messy English. Replace 'induction-fold'.
 - pg8, line 288; 'model cells' → 'cell models'
 - pg8, line 299; it is not clear what you mean by 'basal promoter'
 - pg25, line 757; replace 'shots' ; pg11, line 409 : "mTurquoise2"
- Results
 - pg5, lines 151-154: include more detailed description of how promoters were designed, where the parts came from.
 - Include some measure of burden that the sensors presented in this study have on host cells; this can be growth of cells with fully active signaling and sensing modules; alternatively, it could be the expression levels of a constitutive genomic reporter in cells with fully active signaling and sensing modules.
 - pg8, line 298-299; not clear why you are referring to Supplementary Figure 11 here. Actually, there seems to be an error in the numbering of Supplementary Figures; please correct this.
 - pg8, 'Cross-kingdom universality of the designed channels'; I am interested to see whether the same approach of promoter and strain design would work for the other communication channels that were presented (optional).
 - pg8, line 320; refer to Fig4c,d where you refer to Supp Fig 16.
 - pg9, line 336-340; show some diagram of the genetic circuit used to implement the NOR gates.
- Discussion
 - pg9, line 354-356; please reword the first few sentences of the discussion. Strange English and potential to be misinterpreted or misunderstood.
 - pg9, lines 362-368; missing referencing.
 - Expand referencing in Discussion, possibly targeted to this field's high impact publications.
- Methods
 - Specify where verification by sequencing was performed. Specify how cell lines were verified for integration (optional).
 - pg12, line 480; 'white cell' → 'non-fluorescent cell' replace
 - pg12, line 485; include centrifugation speed and time throughout.
- Figures
 - Some figures have too small labels that are hard to read (e.g. Fig. 2, y-axes).
 - pg24, Fig 3; include quantitative measurements of data presented in the heatmaps as a histogram showing actual values. The values are hard to read off a color map. This can be presented in Supplementary materials.
- Supplementary info
 - Essential: Sequences of all developed genetic components and enzymes has to be available in Supplementary information in detail. A copiable sequence for all genetic tools that were used must be easily available.
 - Much richer information on all utilized parts has to be included.

- All abbreviations have to be explained; e.g. Supp Fig 10 contains many abbreviations that are unexplained or ambiguous.
- Some figures have too small labels that are hard to read (e.g. Supp Fig 11).
- Sequences should be copiable text.
- pg20: some explanation should be provided as to how the Sal system was implemented into *S. cerevisiae*. If it was done analogously to the DAPG-PhIF system, this should be mentioned.

Reviewer #2:

Remarks to the Author:

The authors have described an approach to develop new communication pathways that can be added to bacterial and eukaryotic cells that will enable molecular control over cells and cell networks. In general, the article "hits the spot" in that there is a tremendous need for advancements in this area and just a few innovative studies that have helped to define workable solutions. The methodologies proposed here are rational; the importance is high and the example methodology is noteworthy. My enthusiasm for this work is predicated on the need to solve difficult issues concerning cell networks that exist in nature, particularly those where there is limited analytical access and the need is great. I am thinking of the GI tract, for example. If we are to get a handle on understanding the influence of the GI microbiome on human physiology, we will need to engineer signaling among networks of cells and the community activities should be actuated in non-invasive ways. Approaches such as those provided by the authors, while to me a logical extension of earlier efforts, will be needed.

The experiments appear to be well organized, executed, and with a few grammatical issues, well described. This reviewer is of the opinion that the overall methodology is robust. I have provided some thoughts below that will help with the presentation, justification, and novelty.

Primary thoughts:

The authors have based the work on requirements for (i) universality and (ii) orthogonality. The need for universality seems valid in that the synthesis of signal molecules should be such that exogenous precursors aren't needed, instead typical biosynthetic functions should enable their synthesis. In this way, the signals can be generated from a variety of organisms, in particular, those that will be resident in the niche community of interest. The orthogonality argument has been made by many and is reasonable here. In order to meet these requirements, constraints were imposed, however, that limit the potential application and appeal. These should be either discussed in more detail or otherwise validated.

1. One constraint is that the signal molecules freely diffuse across cell membranes. This dramatically reduces the number of viable signal molecules and also limits the mode by which they act in the host cells (i.e., binding to transcription factors). Was there a robust examination of this limitation? Some additional discussion is warranted. For example, the types of pathways that would be available for signal molecule generation is likely limited. Also, the gene circuits that can be controlled using aTF's will be limited. These constraints might have been buried, intentionally or unintentionally, in the experimental results. For example, the authors validated the orthogonality requirement by demonstrating minimal cross talk - they used the aTF's that were an intrinsic part of the study. There may have been other metabolic consequences that went undetected, but could have ramifications relative to end use. Some discussion is warranted here relative to the number of signal molecules, the types of pathways available for their synthesis and the limiting consequences of using aTFs.

2. The authors have reduced the experimental range of the study by limiting the number of enzymatic steps in the signal synthesis pathway. Specifically, they chose five enzymes as a maximum and justified this by "reducing cellular resource taxing". It would be important to show how this was conceived. Currently, this seems to be an arbitrary decision. Some enzymes for

example, may represent a huge burden to the cells and could be in a two-step pathway, while a 7 step pathway might have minimal consequences based on flux through the pathway and the levels of enzymes needed. Also, they have not provided any data associated with this constraint. If these aren't shown, then there is no need to suggest that it is fundamental to the approach. Rather, it was a convenience.

3. Cross-kingdom universality. Why show this in HEK cells? Is there an intent to develop methodologies for urinary tract infections? It would seem to be more important to show Cross kingdom utility in a relevant cell line, such as an intestinal epithelial line. Provide some additional rationale here.

4. The logic gate circuits. It would be helpful if the authors provide an example where these complex circuits would be useful. The premise of the paper is on developing the molecules and pathways. What computations are performed using these strains in logic gate circuits. This section of the paper is limited to a paragraph and its utility could be strengthened.

Minor concerns (no particular order of importance):

L 49. Short- medium- long-range...perhaps better terms are autocrine, paracrine, endocrine...

L 54. This refers to "molecular communication" might cite this literature, especially as it relates to QS.

L 66. Autonomous induction was first shown in 2010 (Tsao et al, Metabolic Engineering).

L 66. Autonomous population control last year (Stephens et al, Nature Comm.). Both of the above exploit endogenous (AI-2) QS signaling.

L 82. AI-2 is an endogenous interkingdom signaling molecule (Zargar, mBio).

L 91. Missed recent pub on methods to develop orthogonal signals (McClune et al., Nature).

L 91. Presumes that the signaling molecules must pass through cell membranes and find their way to aTFs. This is a considerable challenge, one met by the authors, and also a severe limitation on the approach in that it presumes that the receiving cells have no metabolic activity for the signal molecule (as noted above).

L 104. Some of them...state what it was...three?

L 126. Not sure I agree with the need to limit to 5 enzymes to reduce cellular taxing (as above). The load of the signal generation itself could be distributed among cells. Moreover, if one enzyme represents more of a burden than others, this would preclude this method from working. This is a significant limitation, seemingly arbitrary. Perhaps the method should include an in silico method for evaluating this criterion?

L 150. Two commas in a row.

L 152. Several [of] the basal...

L 170-180. This methodology is not novel. There is no discussion of the background level, just amplification (normalized). If there is appreciable background, this would erroneously be attributed to open communication.

L 187-198. This is a lot of work. Well done. It could alone be the topic of several papers. It is a

standard method, however.

L 190-220. Co-Cultures. How did you stimulate the synthesis of the signal molecule. It would seem to be important to have a working system that opens a communication channel by providing a stimulus. In the co-culture, the stimulus is important, then the time over which the sender cells respond by making the signal molecule, the transport of the signal molecule to the receiver cells (autocrine, paracrine, endocrine) and the subsequent transport into a response of the receiver cells.

L251-L270. While this is a straightforward set of experiments and seems to have been expertly executed, what is the point? Also, did you screen using sequence searches to make sure that there aren't any aTFs in the genomes of the supposed non-responding receivers?

L 289. Why did you transfer "some" and not all 10?

L 299. The transfer to HEK cells is very nice. It should be presented with more data and its own figure.

L308-L350. Interesting work. This demonstrates the orthogonality of the signals and the sender and receiver cells. It is done on plates and depends on the diffusion of the signal molecules. I wonder what the application of this might ever be?

Figures.

2. It is important to see the background levels. Suggest in supplemental providing the FACS data. Specifically, one of the most important issues in studies like this has to do with the diffusion and perception of the signal molecule. The fold induction, if performed using a fluorimeter, would give you the total fluorescence. But, the number of cells might be affected as well as, or instead of, the expression rate of the fluorophore in the receiving cell. Total fluorescence is the sum of both and these are important. Please specify what was used here. If the number of cells that fluoresce is the primary measure, then the transport of the signal molecule might have been a variable (that was not accounted for).

3. I'm not sure why the YFP is normalized by RFP. We need amplification from zero not from a different promoter.

Point-by-point responses to the reviewers.

Questions/comments from Reviewer #1 (related comments may be grouped together):

Question 1: Strengths and weaknesses: Some of the much-needed cell-to-cell communication tools for use in Synthetic Biology are developed in this study. Not only are these tools implemented for use in *E. coli*; some are shown to be functional in yeast and mammalian cells and could, thus, be used for inter- kingdom communication. The study also implements original and exciting spatially segregated logic gate circuitry that has the potential to be of good interest to the Synthetic Biology community.

Response: We thank the reviewer for these positive comments and the many useful suggestions which we have followed as in the responses below.

Question 2: Please provide a complete and comprehensive cell burden analysis; how constructs impact growth in *E. coli* (i.e. provide growth curves).

Response: We have followed the suggestion and measured the growth curves of the sender and receiver strains. All the experimental results of their growth curves are provided in Supplementary Figure 22 and 23, and additional discussion are added to Supplementary Information (Line 367-393). Briefly, none of the ten receiver parts exerted discernible burden on the host cell growth, while some of the sender parts resulted in severe growth defect for their host cells. Especially, for the Sal and DAPG sender systems, the growth of their host cells was dramatically hindered by the burden or toxicity of the biosynthetic enzymes and signal molecules involved.

Question 3: Detailed presentation of all genetic sequences and constructs. Minor, but potentially useful: include characterization of signaling modules in different growth media in *E. coli* (M9, LB, 2xYT).

Response: We added all the sequences of the sender and the receiver parts in the supplementary files. As for robustness to growth conditions, we indeed evaluated the differences between LB and M9 media for several signaling systems. The results below show similar induction performances of four systems in M9 and LB, respectively, indicating that our characterization of the sensitivity and the induction fold are not significantly affected by the growth media.

Question 4: Expand referencing in Discussion.

Response: More references and expanded discussion have been added to the manuscript.

Question 5: Include plots of actual values presented on the heatmaps.

Response: We have added the values of the samples on the heatmap into Supplementary Figures 24 and 25.

Question 6: Native English copy-editing: Minor language corrections throughout; only a small subset of what has to be done is mentioned here. In general, it is OK, but lacks the native English form. Figure 1 legend sounds strange in past tense; pg7, lines 244-onwards; a little messy English. Replace ‘induction-fold’. o pg8, line 288; ‘model cells’ → ‘cell models’; pg8, line 299; it is not clear what you mean by ‘basal promoter’; pg25, line 757; replace ‘shots’; pg11, line 409 : "mTurquoise2";

Response: We are grateful for such helpful comments, and have fixed language issues throughout the text. We have also sent the manuscript to a native English-speaking editing service for polishing.

Question 7: pg5, lines 151-154: include more detailed description of how promoters were designed, where the parts came from.

Response: Detailed description of all the promoter designs are provided in Supplementary information lines 21-49, lines 69-92 and lines 111-150. The origin of all the parts are provided in Supplementary Table 1.

Question 8: Include some measure of burden that the sensors presented in this study have on host cells; this can be growth of cells with fully active signaling and sensing modules; alternatively, it could be the expression levels of a constitutive genomic reporter in cells with fully active signaling and sensing modules.

Response: In order to simplify the added experiments, we measured the growth curves of the host cells for all sender and receiver parts. A comprehensive analysis of cell burden has been provided in Supplementary Information, with the growth curves of senders and receivers shown in Supplementary Figures 22 and 23.

Question 9: pg8, line 298-299; not clear why you are referring to Supplementary Figure 11 here. Actually, there seems to be an error in the numbering of Supplementary Figures; please correct this.

Response: We are sorry for our carelessness in editing and have corrected it and the following Figures.

Question 10: pg8, ‘Cross-kingdom universality of the designed channels’; I am interested to see whether the same approach of promoter and strain design would work for the other communication channels that were presented (optional).

Response: We thank the reviewer for the insightful question. Based on our design rules, we expected all the signal channels to be functional in all species. However, we did notice some sender parts which include multiple enzymes tended to cause burden to the host cells or were hard to be constructed in mammalian cells. In comparison, the signaling systems with simple and burden-free sender parts would be easily transferred. We have thus changed the section title to “Cross-kingdom capability of the designed channels”.

Question 11: pg8, line 320; refer to Fig4c,d where you refer to Supp Fig 16. pg9, line 336-340; show some diagram of the genetic circuit used to implement the NOR gates.

Response: We have provided the diagrams of all NOR gates in Supplementary Figure 20.

Question 12: pg9, line 354-356; please reword the first few sentences of the discussion; Strange English and potential to be misinterpreted or misunderstood. pg9, lines 362-368; missing referencing. Expand referencing in Discussion, possibly targeted to this field’s high impact publications.

Response: We have revised the sentences and expanded the references in the Discussion section.

Question 13: Specify where verification by sequencing was performed. Specify how cell lines were verified for integration (optional). pg12, line 480; ‘white cell’ → ‘non-fluorescent cell’ replace; pg12, line 485; include centrifugation speed and time throughout.

Response: As a standard rule of DNA construction, we have verified by sequencing all the PCR amplified DNA fragments. For the mammalian cell line, we did not sequence them again. For the verification of the integration of DNA in HEK293T cell-line, both transfected sender and receiver modules carried fluorescent proteins, and positive clones were verified using flow cytometer and fluorescent microscopy, respectively.

Question 14: Figures: Some figures have too small labels that are hard to read (e.g. Fig. 2, y-axes). pg24, Fig 3; include quantitative measurements of data presented in the heatmaps as a histogram showing actual values. The values are hard to read off a color map. This can be presented in Supplementary materials.

Response: The small labels have been properly enlarged. The values of heatmap are provided in Supplementary Figures 24 and 25.

Question 15: Supplementary info: Essential Sequences of all developed genetic components and enzymes has to be available in Supplementary information in detail. A copiable sequence for all genetic tools that were used must be easily available.

Response: We have added all the sequences for ten sender and receiver parts of the signaling systems in Supplementary Information.

Question 16: Supplementary info: All abbreviations have to be explained; e.g. Supp Fig 10 contains many abbreviations that are unexplained or ambiguous.

Response: We have added explanations to the legend of Supplementary Figure 10.

Question 17: Supplementary info: some explanation should be provided as to how the Sal system was implemented into *S. cerevisiae*. If it was done analogously to the DAPG-PhIF system, this should be mentioned.

Response: More detailed information of implementing Sal into *S. cerevisiae* system have been added to Supplementary Information (Line 252-257).

Questions/comments from Reviewer #2 (related comments may be grouped together):

Question 1: The authors have described an approach to develop new communication pathways that can be added to bacterial and eukaryotic cells that will enable molecular control over cells and cell networks. In general, the article “hits the spot” in that there is a tremendous need for advancements in this area and just a few innovative studies that have helped to define workable solutions. The methodologies proposed here are rational; the importance is high and the example methodology is noteworthy. My enthusiasm for this work is predicated on the need to solve difficult issues concerning cell networks that exist in nature, particularly those where there is limited analytical access and the need is great. I am thinking of the GI tract, for example. If we are to get a handle on understanding the influence of the GI microbiome on human physiology, we will need to engineer signaling among networks of cells and the community activities should be actuated in non-invasive ways. Approaches such as those provided by the authors, while to me a logical extension of earlier efforts, will be needed. The experiments appear to be well organized, executed, and with a few grammatical issues, well described. This reviewer is of the opinion that the overall methodology is robust. I have provided some thoughts below that will help with the presentation, justification, and novelty.

Response: We thank the reviewer for the nice comments and the insightful suggestion

on the potential applications in the GI microbiome and other fields.

Question 2: The orthogonality argument has been made by many and is reasonable here. In order to meet these requirements, constraints were imposed, however, that limit the potential application and appeal. These should be either discussed in more detail or otherwise validated. 1. One constraint is that the signal molecules freely diffuse across cell membranes. This dramatically reduces the number of viable signal molecules and also limits the mode by which they act in the host cells (i.e., binding to transcription factors). Was there a robust examination of this limitation? Some additional discussion is warranted. For example, the types of pathways that would be available for signal molecule generation is likely limited. 2. Also, the gene circuits that can be controlled using aTF's will be limited. These constraints might have been buried, intentionally or unintentionally, in the experimental results. For example, the authors validated the orthogonality requirement by demonstrating minimal cross talk - they used the aTF's that were an intrinsic part of the study. There may have been other metabolic consequences that went undetected, but could have ramifications relative to end use. Some discussion is warranted here relative to the number of signal molecules, the types of pathways available for their synthesis and the limiting consequences of using aTFs.

Response: We appreciate the reviewer for such insightful comments on the limitations of the orthogonality. The biosynthesis pathways of our signaling molecules were from secondary metabolic pathways that do not tend to occasionally exist in a target species. The KEGG database could also help us identify possible enzymes that may unexpectedly synthesize or consume signal molecules. We thus expected that the engineered cell-cell communication could be orthogonal with any host except the ones they are originated (i.e. Sal-NahR in *Pseudomonas putida*). Indeed, other researchers have also developed cell-cell communication systems based on fungal mating peptide/G-protein-coupled receptor (GPCR) pairs in *S. cerevisiae*. However, these systems have quite good orthogonality, but not universality, because it is difficult to transfer parts from fungi to bacteria and mammalian cells. We have revised the related discussion in the manuscript.

Question 3: The authors have reduced the experimental range of the study by limiting the number of enzymatic steps in the signal synthesis pathway. Specifically, they chose five enzymes as a maximum and justified this by "reducing cellular resource taxing". It would be important to show how this was conceived. Currently, this seems to be an arbitrary decision. Some enzymes for example, may represent a huge burden to the cells and could be in a two-step pathway, while a 7-step pathway might have minimal consequences based on flux through the pathway and the levels of enzymes needed. Also, they have not provided any data associated with this constraint. If these aren't shown, then there is no need to suggest that it is fundamental to the approach. Rather, it was a convenience.

Response: We thank the reviewer for the critical argument. We agree with the

reviewer that it is too arbitrary to state five enzymes being the maximum number to reduce the cellular resource taxing. We have removed the statement from the manuscript. Indeed, by measuring the growth burden of each sender, we found the Sal sender, produced from a two-step pathway, is actually more toxic than the IV sender produced from a five-enzyme pathway, possibly due to the high concentration of the signal molecule or toxic intermediates of the Sal biosynthesis pathway.

Question 4: Cross-kingdom universality. Why show this in HEK cells? Is there an intent to develop methodologies for urinary tract infections? It would seem to be more important to show Cross kingdom utility in a relevant cell line, such as an intestinal epithelial line. Provide some additional rationale here.

Response: Although GI microbiome modulation would be an excellent and immediate application outlet for engineered cross-kingdom communication, we would not limit the system's potential in other biomedical areas, since symbiont microbiome is being discovered in many types of human tissues, with the latest being in adipose and tumor¹. We performed experiments in HEK293T and Hela cell lines (results not shown in the manuscript) because both are long-established model systems and are easy to culture and transfect. The success in these cells are by itself significant progress given prior studies showing it is generally hard to engineer exogenous signaling pathways in mammalian cells versus in microbes. We believe that with or without a few adjustments, the cell-cell communication system would apply to other types of human cells.

Question 5: It would be helpful if the authors provide an example where these complex circuits would be useful. The premise of the paper is on developing the molecules and pathways. What computations are performed using these strains in logic gate circuits. This section of the paper is limited to a paragraph and its utility could be strengthened.

Response: We have added one paragraph about the complex circuits in the discussion section. We believed the distributed logic circuits could be used to construct stable multiple-strain microbial consortia for environmental and human health applications.

Question 6: L49. Short- medium- long-range...perhaps better terms are autocrine, paracrine, endocrine...; L54. This refers to “molecular communication” might cite this literature, especially as it relates to QS.; L66. Autonomous induction was first shown in 2010 (Tsao et al, Metabolic Engineering).; Autonomous population control last year (Stephens et al, Nature Comm.). Both of the above exploit endogenous (AI-2) QS signaling.; L82. AI-2 is an endogenous interkingdom signaling molecule (Zargar, mBio).; L91. Missed recent pub on methods to develop orthogonal signals (McClune et al., Nature).; L 104. Some of them...state what it was...three? L150. Two commas in a row. L152. Several [of] the basal...

Response: We thank the reviewer for pointing out recent progress in literature and the ambiguities in writing. We have added the references into the manuscript and also

¹ Massier L, Chakaroun R, Tabei S, et al. Adipose tissue derived bacteria are associated with inflammation in obesity and type 2 diabetes. *Gut*. 2020;gutjnl-2019-320118.
Nejman D, Livyatan I, Fuks G, et al. The human tumor microbiome is composed of tumor type-specific intracellular bacteria. *Science*. 2020;368(6494):973-980

revised the relevant sentences.

Question 7: L170-180. This methodology is not novel. There is no discussion of the background level, just amplification (normalized). If there is appreciable background, this would erroneously be attributed to open communication.

Response: Yes, the co-cultured strategy is not novel, we thus added a reference to previous work. We have added the background data in Supplementary Figure 24. The background fluorescence level of each induction system depends on both transcriptional and translational activity of the reporter gene. Therefore, we believe the dynamic range is a more important aspect for the system.

Question 8: L187-198. This is a lot of work. Well done. It could alone be the topic of several papers. It is a standard method, however.

Response: We thank the reviewer for the nice comment. We indeed did a lot of optimization on the well-studied QS systems in order to obtain high performant cell-cell communication system. This helped a lot in adapting these systems to mammalian cells and expand the cross-kingdom communication toolbox.

Question 9: L 190-220. Co-Cultures. How did you stimulate the synthesis of the signal molecule. It would seem to be important to have a working system that opens a communication channel by providing a stimulus. In the co-culture, the stimulus is important, then the time over which the sender cells respond by making the signal molecule, the transport of the signal molecule to the receiver cells (autocrine, paracrine, endocrine) and the subsequent transport into a response of the receiver cells.

Response: In this case, the synthesis of signal molecule was controlled by a P_{tet} promoter (Supplementary Figure 4b). The aTc inducer (200ng/ml) was added to the media as stimulus at the beginning of the co-culture. The dose-response curves using synthetic signal molecule are similar to the corresponding co-culture curves (Figure 2c).

Question 10: L251-L270. While this is a straightforward set of experiments and seems to have been expertly executed, what is the point? Also, did you screen using sequence searches to make sure that there aren't any aTFs in the genomes of the supposed non-responding receivers?

Response: These experiments were to show comprehensively the extent to which non-cognate sender and receiver modules could communicate, which form the basis for channel specificity when multiple communication pathways are needed in one application scenario. It is a good question about the possibility of signal molecules stimulating endogenous aTFs in the host genomes. The original organism of all the aTFs are shown in Supplementary Table 1. In *E. coli* and other host cells used in this work, none of the aTFs were found in the genome. Moreover, the fact that the signal molecules in this work were chosen to be structurally diverse was precisely to minimize the chance of crosstalk with potential aTFs encoded in host genomes and any exogenous aTFs that are implanted in host genomes.

Question 11: L299. The transfer to HEK cells is very nice. It should be presented with more data and its own figure. I'm not sure why the YFP is normalized by RFP. We need amplification from zero not from a different promoter.

Response: We thank the reviewer for the nice comment. The RFP (mCherry) protein was constitutively expressed on the plasmid that was transiently transfected. It was used to indicate the copy number of the plasmid in a HEK293T cell. The normalization of YFP to RFP was to eliminate the effect of plasmid copy number variations. The original data are shown below and in Supplementary Figure 12.

Question 12: L308-L350. Interesting work. This demonstrates the orthogonality of the signals and the sender and receiver cells. It is done on plates and depends on the diffusion of the signal molecules. I wonder what the application of this might ever be?

Response: These experiments were plain demonstrations of the robustness and orthogonality of our cell-cell communication systems. The level of robustness and orthogonality required for completing such complex logic have not been met in previous work. We believed that such patterns formed through multi-channel communications would inspire the design of artificial tissues and complex spatially structured microbial consortia.

Question 13: Figures: It is important to see the background levels. Suggest in supplemental providing the FACS data. Specifically, one of the most important issues in studies like this has to do with the diffusion and perception of the signal molecule. The fold induction, if performed using a fluorimeter, would give you the total fluorescence. But, the number of cells might be affected as well as, or instead of, the expression rate of the fluorophore in the receiving cell. Total fluorescence is the sum of both and these are important. Please specify what was used here. If the number of cells that fluoresce is the primary measure, then the transport of the signal molecule might have been a variable (that was not accounted for).

Response: We thank the reviewer for such insightful comment. All the experimental data were collected by flow cytometry. Indeed, the background level of cells is critical for determining the actual fluorescence of the samples. Thus, as described in the material and method section of the manuscript, we subtracted the mean YFP fluorescence of a non-fluorescent negative control from each sample to remove the

background value. The negative control sample was plated the at same time as other samples were and the cells were transfected with an empty vector plasmid (without YFP). To better illustrate the fluorescence of the null control *vs* other samples, we have included some examples of the original FACS data of fluorescent cells and the null control in Supplementary Figure 12. If the reviewer referred to the leaked expression as “background levels”, we have added these values in Supplementary Figure 24.

Reviewers' Comments:

Reviewer #1:

Remarks to the Author:

The authors have addressed all our comments and this paper is suitable for publication.

Reviewer #2:

Remarks to the Author:

The authors have addressed my concerns.